

# Development and validation of web-based dynamic nomograms predictive of disease-free and overall survival in patients who underwent pneumonectomy for primary lung cancer

Xiangyang Yu[1,*], Feng Wang[2,*], Longjun Yang[3], Kai Ma[1],
Xiaotong Guo[1], Lixu Wang[1], Longde Du[1], Xin Yu[1], Shengcheng Lin[1],
Hua Xiao[1], Zhilin Sui[1], Lanjun Zhang[4] and Zhentao Yu[1]

[1] Department of Thoracic Surgery, National Cancer Center/National Clinical Research Center for Cancer/Cancer Hospital & Shenzhen Hospital, Chinese Academy of Medical Sciences and Peking Union Medical College, Shenzhen, Guangdong, China
[2] Department of Minimally Invasive Surgery, Beijing Chest Hospital, Capital Medical University; Beijing Tuberculosis and Thoracic Tumor Research Institute, Beijing, Beijing, China
[3] Department of Thoracic Surgery, Shandong Cancer Hospital and Institute, Shandong First Medical University and Shandong Academy of Medical Sciences, Jinan, Shandong, China
[4] Department of Thoracic Surgery, State Key Laboratory of Oncology in South China, Collaborative Innovation Center for Cancer Medicine, Sun Yat-sen University Cancer Center, Guangzhou, Guangdong, China
* These authors contributed equally to this work.

Corresponding authors
Lanjun Zhang,
zhanglj@sysucc.org.cn
Zhentao Yu, yztao2015@163.com

## ABSTRACT

**Background:** The tumour-node-metastasis (TNM) staging system is insufficient to precisely distinguish the long-term survival of patients who underwent pneumonectomy for primary lung cancer. Therefore, this study sought to identify determinants of disease-free (DFS) and overall survival (OS) for incorporation into web-based dynamic nomograms.

**Methods:** The clinicopathological variables, surgical methods and follow-up information of 1,261 consecutive patients who underwent pneumonectomy for primary lung cancer between January 2008 and December 2018 at Sun Yat-sen University Cancer Center were collected. Nomograms for predicting DFS and OS were built based on the significantly independent predictors identified in the training cohort ($n = 1,009$) and then were tested on the validation cohort ($n = 252$). The concordance index (C-index) and time-independent area under the receiver-operator characteristic curve (AUC) assessed the nomogram's discrimination accuracy. Decision curve analysis (DCA) was applied to evaluate the clinical utility.

**Results:** During a median follow-up time of 40.5 months, disease recurrence and death were observed in 446 (35.4%) and 665 (52.7%) patients in the whole cohort, respectively. In the training cohort, a higher C-reactive protein to albumin ratio, intrapericardial pulmonary artery ligation, lymph node metastasis, and adjuvant therapy were significantly correlated with a higher risk for disease recurrence; similarly, the independent predictors for worse OS were intrapericardial pulmonary artery and vein ligation, higher T stage, lymph node metastasis, and no adjuvant

therapy. In the validation cohort, the integrated DFS and OS nomograms showed well-fitted calibration curves and yielded good discrimination powers with C-index of 0.667 (95% confidence intervals CIs [0.610–0.724]) and 0.697 (95% CIs [0.649–0.745]), respectively. Moreover, the AUCs for 1-year, 3-year, and 5-year DFS were 0.655, 0.726, and 0.735, respectively, and those for 3-year, 5-year, and 10-year OS were 0.741, 0.765, and 0.709, respectively. DCA demonstrated that our nomograms could bring more net benefit than the TNM staging system.

**Conclusions:** Although pneumonectomy for primary lung cancer has brought encouraging long-term outcomes, the constructed prediction models could assist in precisely identifying patients at high risk and developing personalized treatment strategies to further improve survival.

# INTRODUCTION

A multimodality treatment strategy with pneumonectomy is reserved for lung cancer patients in whom tumours invade the main bronchus, pulmonary vessels, or ipsilateral lobe(s) to achieve prolonged survival (*Ettinger et al., 2021*; *Yu et al., 2021*). Although reports based on the Surveillance, Epidemiology, and End Results (SEER) program and the National Cancer Database (NCDB) show an apparent decrease in the proportion of pneumonectomies for the surgical treatment of lung cancer over the past 20 years, the constitute ratio remains at 5–10% (*Hancock et al., 2014*; *Yu et al., 2021*). However, due to the high perioperative morbidity and mortality, reduced lung function, and poor quality-of-life, the long-term survival rate after pneumonectomy remains unsatisfactory and varies (10.8–66.0%) (*Dickhoff et al., 2016*; *Hancock et al., 2014*; *Herskovic et al., 2017*; *Yu et al., 2021*). Despite many studies that have identified predictive factors for postpneumonectomy survival, the ultimate aims of these studies were principally limited to the confirmation of an isolated clinicopathological feature as a predictor (*Rivera et al., 2014*; *Tabutin et al., 2012*). The anatomy-based tumour-node-metastasis (TNM) staging is the most widely used model for risk stratification and survival prediction of patients with lung cancer; nevertheless, recent studies have demonstrated that the current staging system is not sufficient to precisely distinguish the long-term survival of patients who underwent pneumonectomy for lung cancer (*Dickhoff et al., 2016*; *Herskovic et al., 2017*; *Tabutin et al., 2012*). Therefore, individualized postpneumoneuctomy management continues to be a challenge for thoracic surgeons.

A nomogram is a pictorial depiction that can be used to generate a numerical risk probability of a specific clinical outcome (such as complication, recurrence, and death) (*Balachandran et al., 2015*). Therefore, the clinical use of a nomogram that is tailored to the determinate prognostic variables of an individual patient can facilitate a personalized follow-up schedule and a multimodality treatment strategy in oncology (*Liu et al., 2020*). In addition, a web-based calculator that is transferred from the nomogram can provide a more intuitive and convenient interface to assist in communication with patients

(*Amar et al., 2019*). Although several nomograms to predict the prognosis of patients with lung cancer were established in previous studies, an exclusive and online nomogram for this specific population with removal of an entire lung has not yet been available in clinical practice.

Therefore, we performed the present study based on a real-world cohort analysis with the aim of identifying the independent clinicopathological variables that predict disease-free and overall survival (DFS and OS) in patients who underwent pneumonectomy for primary pulmonary malignancy. Moreover, web-based servers according to the integrated nomogram models have been developed and are freely available for thoracic surgeons to input the predictive variables required for the individualized DFS and OS probability.

## MATERIALS AND METHODS

### Patient population

Clinical, pathological, surgical, and follow-up information of patients who underwent one-sided pneumonectomy with curative intent for primary lung cancer between January 2008 and December 2018 was extracted from the medical records at Sun Yat-sen University Cancer Center. Pathological TNM staging was reclassified on the basis of the American Joint Committee on Cancer (AJCC) TNM Staging Manual, 8th Edition. Patients who were less than 18 years of age at surgery, who had other malignant tumours diagnosed before or after pneumonectomy, positive margins (defined as microscopic or macroscopic residual tumor in bronchial stump), who had follow-up times less than 1 month, and who had unknown surveillance outcomes were excluded. A total of 1,261 consecutive patients were enrolled in this study (Fig. S1). Then, by generating a random seed (131) with the aid of the R package "Caret", the patients were divided into a training cohort ($n = 1,009$) and a validation cohort ($n = 252$) at a proportion of 8:2.

### Follow-up strategy and endpoints

After undergoing pneumonectomy or completing adjuvant therapy, the patients were routinely followed up with chest computed tomography (CT) enhancement scans plus cervical and abdominal ultrasonography every 3–6 months in the first 3 years, every 6 months at the 4th to 5th years, and annually thereafter until recurrence or death occurred. To reduce the missing data rate, telephone, letter, or e-mail consultations and smartphone application surveys were selected as supplements for the outpatient follow-up.
All intrathoracic and/or regional recurrence was confirmed by pathology. Distant metastasis was generally diagnosed based on radiology, such as CT, magnetic resonance imaging (MRI), single photon emission computed tomography (SPECT), or positron emission tomography (PET).

DFS time was calculated from the date of pneumonectomy to the date of disease recurrence, death from noncancer causes, or the last date of follow-up (December 31, 2020). OS time was defined as the interval between pneumonectomy and death from any cause or the last follow-up. The patients who had not met the abovementioned endpoints were recorded as censored cases.

## Ethics statement

The institutional ethics committee at Sun Yat-sen University Cancer Center approved this retrospective study (No. B2022-011-01). All patients signed informed consent before surgery.

## Statistical analysis

To analyse the differences between the training and validation cohorts, categorical variables were compared using the $\chi^2$ test or Fisher's exact test, and continuous variables were compared using Student's t test or the Mann-Whitney $U$ test, as appropriate. For the survival analyses, based on the best cut-off values generated by X-tile software (Version 3.6.1; Yale University, New Haven, CT, USA), the continuous variables of neutrophil to lymphocyte ratio (NLR), platelet to lymphocyte ratio (PLR), and C-reactive protein to albumin ratio (CAR) were transformed into categorical variables. The Kaplan-Meier method was used to screen the priori predictors that were significantly associated with DFS and OS in the training cohort. Then, the above significant variables were entered into the least absolute shrinkage and selection operator (LASSO) regression model to further select the most useful prognostic variables with the aid of the minimum lambda ($\lambda$). The R package "glmnet" was used to perform the LASSO regression. Subsequently, the clinicopathological variables selected by the LASSO regression were retained in the final Cox proportional hazards model to determine the independent predictors for survival. All statistical analyses were performed by Statistical Product and Service Solutions (SPSS) version 23.0 software (SPSS; IBM, Inc., Chicago, IL, USA), and a $P$ value less than 0.05 was defined as a significant difference.

The independent predictors for DFS and OS that were identified by the aforementioned Cox proportional hazards models in the training cohort were included to generate two nomograms that were formulated using the R package "rms". A calibration plot was used to estimate the calibration between the actual survival probability and the nomogram estimated survival probability. The discrimination was assessed using the area under the curve (AUC) of a receiver-operator characteristic (ROC) curve.

# RESULTS

## Patient characteristics

In total, 1,370 consecutive patients who underwent one-sided whole lung removal for primary lung cancer between 2008 and 2018 were identified, and 1,261 patients met the inclusion criteria and were included in this study. Although the annual cases of pneumonectomy remained stable over the past 10 years (median: 121, range: 94–129), the constitute ratio of pneumonectomy in the surgical treatment of lung cancer decreased steadily from 13.4% in 2008 to 2.5% in 2018 ($P < 0.001$).

The patients' median age at diagnosis was 57.4 years (range 20–77 years), and most of the patients (86.4%) were male. In all, 131 patients (10.4%) were treated with induction therapy, including one with radiotherapy alone, eight with concurrent chemoradiotherapy, 10 with immunotherapy alone, and 112 with chemotherapy alone. In addition, postoperative adjuvant therapy was immunotherapy alone in three patients, targeted

therapy in four patients, radiotherapy alone in 15 patients, concurrent chemoradiotherapy in 45 patients, and chemotherapy alone in 530 patients. The majority of the pneumonectomies (71.1%) were due to primary squamous cell carcinoma, followed by adenocarcinoma (17.5%), neuroendocrine tumour (4.6%), and adenosquamous carcinoma (2.5%). A minority (18.5%) of the primary tumours were located in the right lung. Pneumonectomy *via* an open approach, three-port video-assisted thoracoscopic surgery (VATS) and uniportal VATS was performed in 1,225 patients (97.1%), 19 (1.5%) and 17 patients (1.3%), respectively. According to the postoperative pathological examination, the constituent ratios of stage 0–I, II, IIIA and IIIB–IV were 9.7%, 33.7%, 38.6% and 18.0%, respectively.

Table 1 lists that characteristics of the training and validation cohort patients were similar.

## Follow-up results

Twenty-seven patients (2.1%) experienced nononcologic mortality within 90 days after the operation, and the 30-day mortality was 1.4% (18 patients). With a median follow-up time of 40.5 months (range: 1.0–153.1 months), tumour recurrence was observed in 446 patients, and 665 patients experienced death events. The estimated median OS and DFS times for all the patients were 60.9 and 77.6 months, respectively. Moreover, the 5-year OS and DFS rates between the training and validation cohorts were not different (52.1% *vs* 49.5%, $P = 0.893$; and 66.2% *vs* 65.9%, $P = 0.821$; respectively).

## Risk factors and predictive nomogram for disease-free survival

According to the univariate analysis for DFS in the training cohort (Table 2), higher CAR (*vs* ≤0.01; $P < 0.001$), intrapericardial pulmonary artery or vein disconnection (*vs* extrapericardium; all $P < 0.001$), adenocarcinoma (*vs* squamous cell carcinoma, SCC; $P < 0.001$), higher T stage ($P = 0.008$), lymph node metastasis (*vs* N0; $P < 0.001$), and adjuvant therapy (*vs* no adjuvant therapy; $P < 0.001$) all correlated with a higher risk for disease recurrence. After LASSO-Cox regression to further reduce possible redundancy, CAR, pulmonary artery disconnection, N stage, and adjuvant therapy remained significant indicators of DFS (Table 3) and were used to build the final nomogram model (Fig. 1A).

Calibration was depicted by drawing the plots of the predicted 1-year, 3-year, and 5-year DFS rates with the confidence intervals (CIs) from the nomogram *vs* the actual probabilities in the training (Figs. 2A–2C) and validation cohorts (Figs. 2D–2F). The nomogram showed good predictive discrimination with a concordance index (c-index) of 0.647 (95% CIs [0.632–0.662]) in the training cohort and 0.667 (95% CIs [0.610–0.724]) in the validation cohort, respectively. Additionally, the AUCs for the 1-year, 3-year, and 5-year DFS generated *via* bootstrap resampling were 0.659, 0.685, and 0.694, respectively, denoting that the model was not overfitted (Figs. 3A–3C). The external validation for the 1-year, 3-year, and 5-year DFS showed AUCs of 0.655, 0.726, and 0.735, respectively, demonstrating the model's good discrimination (Figs. 3D–3F).

For clinical utility, the decision curve analysis (DCA) indicated that using the nomogram model to predict 1-year, 3-year, and 5-year DFS added more net benefit across

**Table 1 Clinical, pathological, and surgical characteristics of patient who underwent pneumonectomy for primary lung cancer in the overall, training, and validation cohorts.**

| Characteristic | Overall (N = 1,261) | Training cohort (N = 1,009) | Validation cohort (N = 252) | P value |
|---|---|---|---|---|
| **Year of operation, n (%)** | | | | |
| 2008–2011 | 455 (36.0%) | 356 (35.3%) | 99 (39.3%) | 0.478 |
| 2012–2015 | 466 (37.0%) | 376 (37.3%) | 90 (35.7%) | |
| 2016–2018 | 340 (27.0%) | 277 (27.5%) | 63 (25.0%) | |
| **Mean age at diagnosis, years (range)** | 57.4 (20–77) | 57.4 (20–77) | 57.3 (–) | 0.859 |
| **Sex, n (%)** | | | | |
| Female | 171 (13.6%) | 140 (13.9%) | 31 (12.3%) | 0.514 |
| Male | 1,090 (86.4%) | 869 (86.1%) | 221 (87.7%) | |
| **Mean duration of chief complaint, months (range)** | 3.3 (0.08–48.0) | 3.3 (0.08–48.0) | 3.2 (0.1–36.0) | 0.707 |
| **Smoking history, n (%)** | | | | |
| Yes | 1,028 (81.5%) | 823 (81.6%) | 205 (81.3%) | 0.937 |
| No | 233 (18.5%) | 186 (18.4%) | 47 (18.7%) | |
| **Mean BMI, kg/m² (range)** | 23.8 (14.2–36.5) | 23.7 (14.2–36.5) | 24.2 (–) | 0.028 |
| **Median weight loss in preoperative 3 months, kg (range)** | 0.0 (0.0–15.0) | 0.0 (0.0–15.0) | 0.0 (0.0–10.0) | 0.123 |
| **Induction therapy, n (%)** | | | | |
| Yes | 131 (10.4%) | 104 (10.3%) | 27 (10.7%) | 0.850 |
| No | 1,130 (89.6%) | 905 (89.7%) | 225 (89.3%) | |
| **Mean FEV1, (range)** | 2.21 (0.77–3.99) | 2.21 (0.77–3.98) | 2.20 (0.92–3.99) | 0.692 |
| **Mean FEV1 % pred, % (range)** | 72.4 (26.7–127.6) | 72.7 (26.7–118.0) | 71.4 (30.0–127.6) | 0.286 |
| **Mean DLCO, (range)** | 7.07 (1.33–32.88) | 7.05 (1.33–17.39) | 7.16 (2.38–32.88) | 0.465 |
| **Mean DLCO % pred, % (range)** | 78.0 (1.4–359.0) | 77.9 (1.4–197.0) | 78.5 (31.0–359.0) | 0.711 |
| **Median NLR, (range)** | 2.43 (0.25–65.87) | 2.46 (0.25–65.87) | 2.71 (0.59–11.17) | 0.004 |
| **Mean PLR, (range)** | 148.30 (16.35–1,264.52) | 149.03 (16.35–1,264.52) | 145.47 (50.53–500.00) | 0.510 |
| **Median CAR, (range)** | 0.00 (0.00–0.93) | 0.00 (0.00–0.84) | 0.00 (0.00–0.93) | 0.338 |
| **Incision method, n (%)** | | | | |
| Open | 1,225 (97.1%) | 978 (96.9%) | 247 (98.0%) | 0.564 |
| Uniportal VATS | 17 (1.3%) | 14 (1.4%) | 3 (1.2%) | |
| Three-portal VATS | 19 (1.5%) | 17 (1.7%) | 2 (0.8%) | |
| **Disconnection of pulmonary artery, n (%)** | | | | |
| Intrapericardium | 180 (14.3%) | 144 (14.3%) | 36 (14.3%) | 0.995 |
| Extra-pericardium | 1,081 (85.7%) | 865 (85.7%) | 216 (85.7%) | |
| **Disconnection of pulmonary vein, n (%)** | | | | |
| Intrapericardium | 226 (17.9%) | 181 (17.9%) | 45 (17.9%) | 0.976 |
| Extra-pericardium | 1,035 (82.1%) | 828 (82.1%) | 207 (82.1%) | |
| **Disconnection of main bronchus, n (%)** | | | | |
| Stapler | 1,238 (98.2%) | 992 (98.3%) | 246 (97.6%) | 0.460 |
| Manual suture | 23 (1.8%) | 17 (1.7%) | 6 (2.4%) | |

| Characteristic | Overall (N = 1,261) | Training cohort (N = 1,009) | Validation cohort (N = 252) | P value |
|---|---|---|---|---|
| **Pathology, n (%)** | | | | |
| Squamous cell carcinoma | 897 (71.1%) | 722 (71.6%) | 175 (69.4%) | 0.970 |
| Adenocarcinoma | 221 (17.5%) | 175 (17.3%) | 46 (18.3%) | |
| Adenosquamous carcinoma | 31 (2.5%) | 24 (2.4%) | 7 (2.8%) | |
| Neuroendocrine tumor | 58 (4.6%) | 46 (4.6%) | 12 (4.8%) | |
| Other | 54 (4.3%) | 42 (4.2%) | 12 (4.8%) | |
| **Grade, n (%)** | | | | |
| Well | 70 (5.6%) | 52 (5.2%) | 18 (7.1%) | 0.451 |
| Moderately | 632 (50.1%) | 506 (50.1%) | 126 (50.0%) | |
| Poorly | 559 (44.3%) | 451 (44.7%) | 108 (42.9%) | |
| **Mean tumor size, cm (range)** | 4.7 (0.0–22.0) | 4.7 (0.2–22.0) | 4.6 (0.0–15.0) | 0.447 |
| **Pathological T stage, n (%)** | | | | |
| T0-1 | 100 (7.9%) | 83 (8.2%) | 17 (6.7%) | 0.580 |
| T2 | 562 (44.6%) | 449 (44.5%) | 113 (44.8%) | |
| T3 | 406 (32.2%) | 318 (31.5%) | 88 (34.9%) | |
| T4 | 193 (15.3%) | 159 (15.8%) | 34 (13.5%) | |
| **Pathological N stage, n (%)** | | | | |
| N0 | 262 (20.8%) | 205 (20.3%) | 57 (22.6%) | 0.243 |
| N1 | 580 (46.0%) | 476 (47.2%) | 104 (41.3%) | |
| N2 | 419 (33.2%) | 328 (32.5%) | 91 (36.1%) | |
| **Pathological TNM stage, n (%)** | | | | |
| 0–I | 122 (9.7%) | 97 (9.6%) | 25 (9.9%) | 0.917 |
| II | 425 (33.7%) | 342 (33.9%) | 83 (32.9%) | |
| IIIA | 487 (38.6%) | 392 (38.9%) | 95 (37.7%) | |
| IIIB–IV | 227 (18.0%) | 178 (17.6%) | 49 (19.4%) | |
| **Adjuvant therapy, n (%)** | | | | |
| Yes | 597 (47.3%) | 465 (46.1%) | 132 (52.4%) | 0.073 |
| No | 664 (52.7%) | 544 (53.9%) | 120 (47.6%) | |
| **Laterality, n (%)** | | | | |
| Right | 233 (18.5%) | 193 (19.1%) | 40 (15.9%) | 0.234 |
| Left | 1,028 (81.5%) | 816 (80.9%) | 212 (84.1%) | |
| **30-day mortality, n (%)** | 18 (1.4%) | 12 (1.2%) | 6 (2.4%) | 0.154 |
| **90-day mortality, n (%)** | 27 (2.1%) | 20 (2.0%) | 7 (2.8%) | 0.435 |
| **5-year DFS rate (%)** | 66.1% | 66.2% | 65.9% | 0.821 |
| **5-year OS rate (%)** | 51.6% | 52.1% | 49.5% | 0.893 |

**Note:**

BMI, body mass index; FEV1, forced expiratory volume in 1 second; DLCO, carbon monoxide diffusing capacity; NLR, neutrophil to lymphocyte ratio; PLR, platelet to lymphocyte ratio; CAR, C-reactive protein to albumin ratio; VATS, video-assisted thoracoscopic surgery; TNM, tumour-node-metastasis; DFS, disease-free survival; OS, overall survival.

**Table 2** The Kaplan-Meier analyses of prognostic predictors of disease-free survival and overall survival.

| Characteristic | No. (%) | Disease-free survival | | Overall survival | |
|---|---|---|---|---|---|
| | | 5-year DFS rate (%) | P value | 5-year OS rate (%) | P value |
| **Age at diagnosis, years** | | | | | |
| <60 | 584 (57.9%) | 60.2% | 0.402 | 53.3% | 0.419 |
| ≥60 | 425 (42.1%) | 57.0% | | 50.5% | |
| **Sex** | | | | | |
| Female | 140 (13.9%) | 53.7% | 0.246 | 52.4% | 0.719 |
| Male | 869 (86.1%) | 59.8% | | 52.1% | |
| **Duration of chief complaint, months** | | | | | |
| ≤1 | 400 (39.6%) | 57.6% | 0.700 | 53.4% | 0.059 |
| 1–3 | 321 (31.8%) | 59.4% | | 48.3% | |
| >3 | 288 (28.6%) | 60.2% | | 54.9% | |
| **Smoking history** | | | | | |
| Yes | 823 (81.6%) | 59.6% | 0.234 | 52.3% | 0.710 |
| No | 186 (18.4%) | 56.3% | | 51.3% | |
| **BMI, kg/m$^2$** | | | | | |
| ≤18.4 | 37 (3.7%) | 47.4% | 0.710 | 44.6% | 0.141 |
| 18.5–23.9 | 519 (51.4%) | 60.9% | | 51.9% | |
| ≥24.0 | 453 (44.9%) | 57.5% | | 53.1% | |
| **Weight loss in preoperative 3 months** | | | | | |
| Yes | 174 (17.2%) | 57.2% | 0.548 | 42.5% | 0.030 |
| No | 835 (82.8%) | 59.3% | | 54.1% | |
| **Induction therapy** | | | | | |
| Yes | 104 (10.3%) | 53.7% | 0.233 | 48.7% | 0.418 |
| No | 905 (89.7%) | 59.6% | | 52.4% | |
| **FEV1 % pred** | | | | | |
| ≥80% | 329 (32.6%) | 59.6% | 0.895 | 55.9% | 0.043 |
| 50–79% | 592 (58.7%) | 59.1% | | 51.5% | |
| <50% | 88 (8.7%) | 54.1% | | 40.9% | |
| **DLCO % pred** | | | | | |
| ≥80% | 422 (41.8%) | 60.3% | 0.802 | 56.7% | 0.008 |
| 60–79% | 402 (39.8%) | 58.2% | | 51.7% | |
| <60% | 185 (18.4%) | 57.7% | | 43.3% | |
| **NLR** | | | | | |
| ≤4.28 | 857 (84.9%) | 59.4% | 0.250 | 53.9% | 0.003 |
| >4.28 | 152 (15.1%) | 56.2% | | 40.9% | |
| **PLR** | | | | | |
| ≤226.85 | 909 (90.1%) | 59.3% | 0.235 | 53.9% | <0.001 |
| >226.85 | 100 (9.9%) | 55.2% | | 33.7% | |
| **CAR** | | | | | |
| ≤0.01 | 568 (56.3%) | 63.2% | <0.001 | 56.8% | 0.005 |
| >0.01 | 441 (43.7%) | 53.6% | | 45.4% | |

| Characteristic | No. (%) | Disease-free survival | | Overall survival | |
|---|---|---|---|---|---|
| | | 5-year DFS rate (%) | *P* value | 5-year OS rate (%) | *P* value |
| **Incision method** | | | | | |
| Open | 978 (96.9%) | 58.5% | 0.273 | 51.3% | 0.017 |
| Uniportal VATS | 14 (1.4%) | 85.7% | | 92.9% | |
| Three-portal VATS | 17 (1.7%) | 63.7% | | 75.5% | |
| **Disconnection of pulmonary artery** | | | | | |
| Intrapericardium | 144 (14.3%) | 41.6% | <0.001 | 29.3% | <0.001 |
| Extra-pericardium | 865 (85.7%) | 61.5% | | 56.1% | |
| **Disconnection of pulmonary vein** | | | | | |
| Intrapericardium | 181 (17.9%) | 44.3% | <0.001 | 31.9% | <0.001 |
| Extra-pericardium | 828 (82.1%) | 61.7% | | 56.6% | |
| **Disconnection of main bronchus** | | | | | |
| Stapler | 992 (98.3%) | 59.0% | 0.439 | 52.2% | 0.289 |
| Manual suture | 17 (1.7%) | 55.8% | | 48.4% | |
| **Pathology** | | | | | |
| Squamous cell carcinoma | 722 (71.6%) | 62.7% | <0.001 | 54.5% | 0.017 |
| Adenocarcinoma | 175 (17.3%) | 42.7% | | 48.0% | |
| Other | 112 (11.1%) | 59.9% | | 43.4% | |
| **Grade** | | | | | |
| Well | 52 (5.2%) | 71.2% | 0.093 | 70.0% | <0.001 |
| Moderately | 506 (50.1%) | 60.2% | | 54.7% | |
| Poorly | 451 (44.7%) | 55.9% | | 47.1% | |
| **Tumor size, cm** | | | | | |
| ≤3 | 195 (%) | 61.5% | 0.074 | 70.0% | <0.001 |
| >3, ≤5 | 486 (%) | 61.1% | | 54.0% | |
| >5, ≤7 | 219 (%) | 53.7% | | 38.0% | |
| >7 | 109 (%) | 54.6% | | 41.2% | |
| **Pathological T stage** | | | | | |
| T0-1 | 83 (8.2%) | 72.8% | 0.008 | 84.6% | <0.001 |
| T2 | 449 (44.5%) | 60.7% | | 58.1% | |
| T3 | 318 (31.5%) | 53.5% | | 41.7% | |
| T4 | 159 (15.8%) | 56.9% | | 41.2% | |
| **Pathological N stage** | | | | | |
| N0 | 205 (20.3%) | 72.0% | <0.001 | 73.3% | <0.001 |
| N1 | 476 (47.2%) | 64.0% | | 57.5% | |
| N2 | 328 (32.5%) | 41.1% | | 31.6% | |
| **Pathological TNM stage** | | | | | |
| 0–I | 97 (9.6%) | 70.8% | <0.001 | 83.7% | <0.001 |
| II | 342 (33.9%) | 67.5% | | 65.8% | |
| IIIA | 392 (38.9%) | 56.3% | | 46.4% | |
| IIIB–IV | 178 (17.6%) | 38.1% | | 23.8% | |

(Continued)

| Characteristic | No. (%) | Disease-free survival | | Overall survival | |
|---|---|---|---|---|---|
| | | 5-year DFS rate (%) | P value | 5-year OS rate (%) | P value |
| **Adjuvant therapy** | | | | | |
| No | 544 (53.9%) | 68.3% | <0.001 | 48.5% | 0.016 |
| Yes | 465 (46.1%) | 49.0% | | 56.9% | |
| **Laterality** | | | | | |
| Right | 193 (19.1%) | 59.4% | 0.832 | 48.7% | 0.082 |
| Left | 816 (80.9%) | 58.9% | | 52.9% | |

**Note:**
DFS, disease-free survival; OS, overall survival; BMI, body mass index; FEV1, forced expiratory volume in 1 second; DLCO, carbon monoxide diffusing capacity; NLR, neutrophil to lymphocyte ratio; PLR, platelet to lymphocyte ratio; CAR, C-reactive protein to albumin ratio; VATS, video-assisted thoracoscopic surgery; TNM, tumour-node-metastasis.

**Table 3 The least absolute shrinkage and selection operator (LASSO)-Cox regression model to further select prognostic predictor of disease-free survival and overall survival.**

| Characteristic | Disease-free survival | | Overall survival | |
|---|---|---|---|---|
| | HR [95% CIs] | P value | HR [95% CIs] | P value |
| **Weight loss in preoperative 3 months** | | | | |
| No | — | — | Reference | |
| Yes | — | — | 1.183 [0.948–1.477] | 0.137 |
| **FEV1 % pred** | | | | |
| ≥80% | — | — | Reference | |
| 50–79% | — | — | 1.196 [0.977–1.464] | 0.082 |
| <50% | — | — | 1.228 [0.865–1.742] | 0.251 |
| **DLCO % pred** | | | | |
| ≥80% | — | — | Reference | |
| 60–79% | — | — | 0.949 [0.774–1.163] | 0.612 |
| <60% | — | — | 1.107 [0.859–1.427] | 0.430 |
| **NLR** | | | | |
| ≤4.28 | — | — | Reference | |
| >4.28 | — | — | 1.129 [0.860–1.482] | 0.383 |
| **PLR** | | | | |
| ≤226.85 | — | — | Reference | |
| >226.85 | — | — | 1.255 [0.917–1.720] | 0.156 |
| **CAR** | | | | |
| ≤0.01 | Reference | | Reference | |
| >0.01 | 1.532 [1.237–1.897] | <0.001 | 1.186 [0.982–1.432] | 0.077 |
| **Disconnection of pulmonary artery** | | | | |
| Pericardium | Reference | | Reference | |
| Extra-pericardium | 0.633 [0.483–0.828] | 0.001 | 0.717 [0.549–0.937] | 0.015 |

| Characteristic | Disease-free survival | | Overall survival | |
|---|---|---|---|---|
| | HR [95% CIs] | *P* value | HR [95% CIs] | *P* value |
| **Disconnection of pulmonary vein** | | | | |
| Pericardium | Reference | | Reference | |
| Extra-pericardium | 0.772 [0.564–1.055] | 0.104 | 0.747 [0.581–0.960] | 0.023 |
| **Pathology** | | | | |
| Squamous cell carcinoma | Reference | | Reference | |
| Adenocarcinoma | 1.534 [1.175–2.003] | 0.002 | 1.120 [0.880–1.426] | 0.356 |
| Other | 1.182 [0.837–1.671] | 0.342 | 1.285 [0.981–1.683] | 0.068 |
| **Grade** | | | | |
| Well | — | — | Reference | |
| Moderately | — | — | 1.517 [0.923–2.493] | 0.100 |
| Poorly | — | — | 1.708 [1.036–2.815] | 0.036 |
| **Pathological T stage** | | | | |
| T0–1 | Reference | | Reference | |
| T2 | 1.401 [0.873–2.247] | 0.162 | 2.339 [1.356–4.035] | 0.002 |
| T3 | 1.681 [1.042–2.713] | 0.033 | 3.239 [1.874–5.598] | <0.001 |
| T4 | 1.343 [0.800–2.254] | 0.264 | 3.108 [1.750–5.519] | <0.001 |
| **Pathological N stage** | | | | |
| N0 | Reference | | Reference | |
| N1 | 1.262 [0.998–1.747] | 0.051 | 1.679 [1.275–2.212] | <0.001 |
| N2 | 2.046 [1.467–2.852] | <0.001 | 3.264 [2.462–4.327] | <0.001 |
| **Adjuvant therapy** | | | | |
| No | Reference | | Reference | |
| Yes | 1.455 [1.168–1.811] | 0.001 | 0.592 [0.494–0.709] | <0.001 |

Note:
HR, hazard ratio; CIs, confidence intervals; FEV1, forced expiratory volume in 1 second; DLCO, carbon monoxide diffusing capacity; NLR, neutrophil to lymphocyte ratio; PLR, platelet to lymphocyte ratio; CAR, C-reactive protein to albumin ratio.

the reasonable threshold probabilities than the TNM stage in both the training (Figs. 4A–4C) and validation cohorts (Figs. 4D–4F).

## Risk factors and predictive nomogram for overall survival

Similarly, through the univariate analysis (Table 2) and the LASSO-Cox proportional hazards regression model, intrapericardial pulmonary artery disconnection, intrapericardial pulmonary vein disconnection, higher T stage, lymph node metastasis, and no adjuvant therapy were identified as independent risk factors for OS (Table 3). The calibration curves between the predicted probability of 3-year, 5-year, and 10-year OS and the actual probability also appeared to have excellent consistency (Fig. S2). The OS nomogram had a C-index of 0.675 ± 0.025 in the training cohort and 0.697 ± 0.048 in the validation cohort, reflecting good discrimination. Furthermore, time-dependent ROCs and AUCs at 3 years, 5 years, and 10 years were used to validate the prognostic accuracy of the

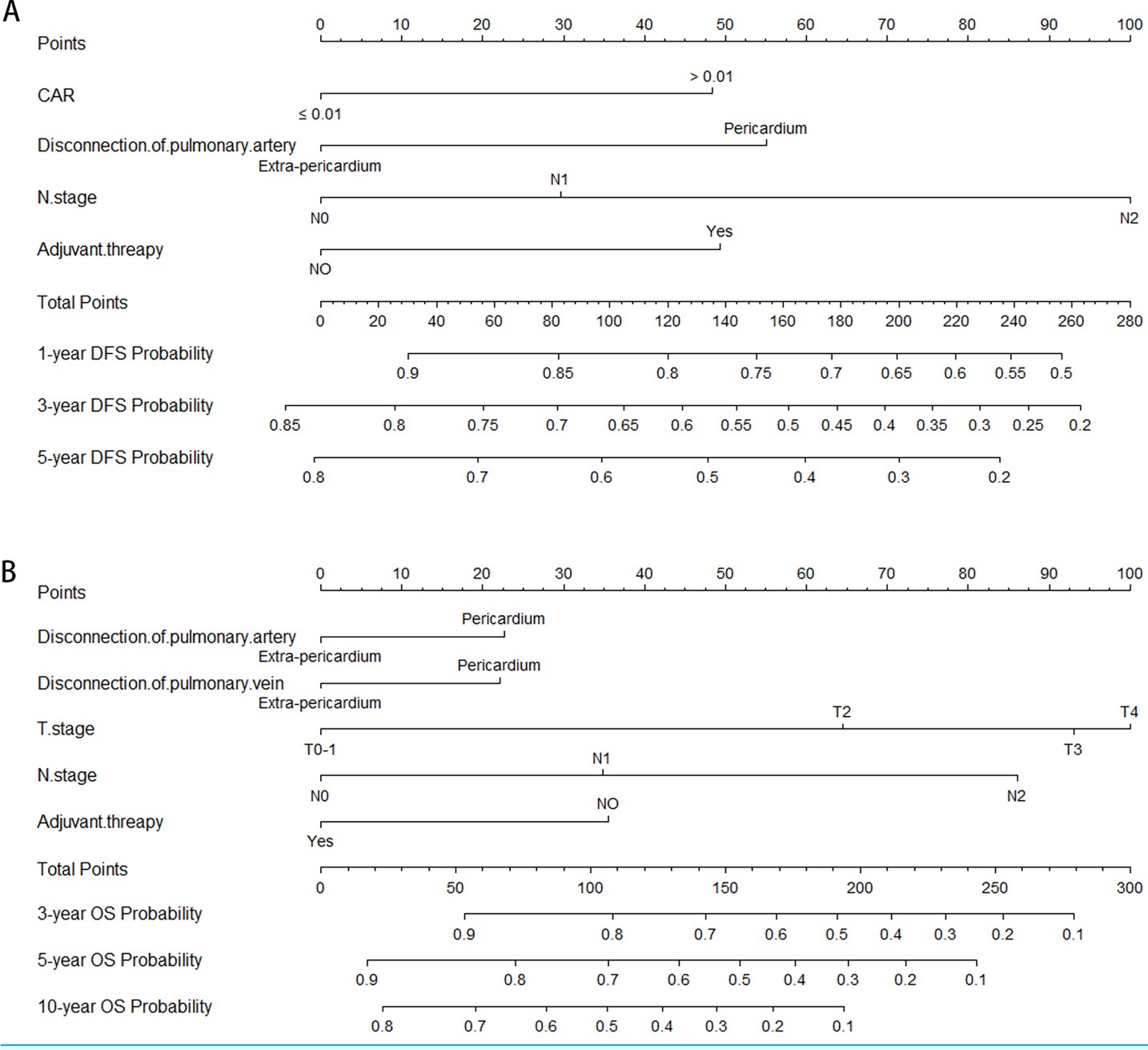

**Figure 1  Nomogram for predicting DFS and OS among patients who underwent pneumonectomy for primary lung cancer.** (A) The sum of the prognostic factor points corresponds to the DFS probability at 1-year, 3-year, and 5-year. (B) The sum of the prognostic factor points corresponds to the OS probability at 3-year, 5-year, and 10-year. CAR, C-reactive protein to albumin ratio.     

OS nomogram (Fig. S3). The DCA indicated that when the threshold probability of a patient or surgeon was greater than 20%, using our OS nomogram to predict the 3-year, 5-year, and 10-year OS could increase the positive benefit more than either the "treat all" scheme, the "treat none" scheme, or the traditional staging system in the training (Figs. S4A–S4C) and validation (Figs. S4D–S4F) groups.

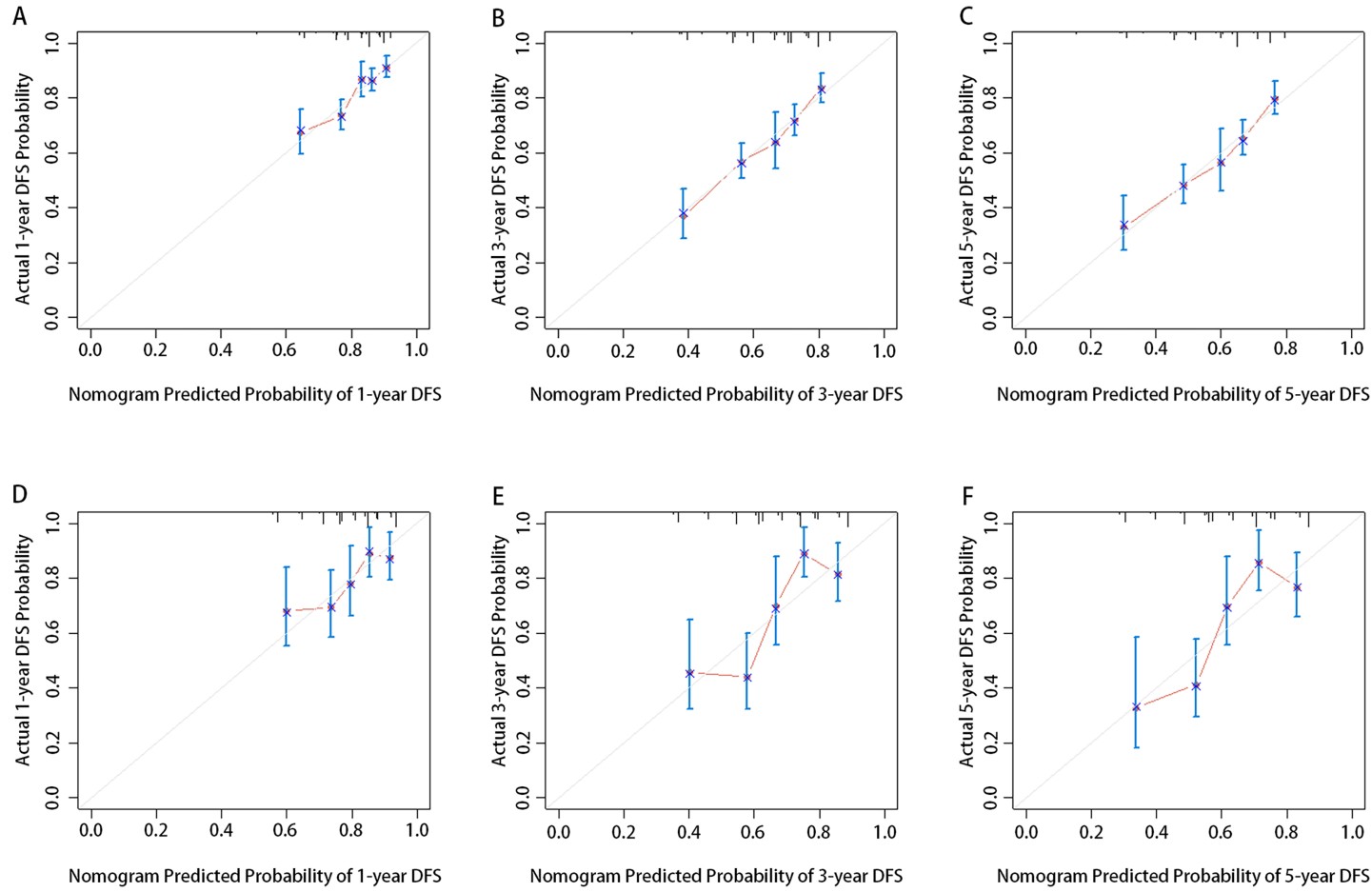

**Figure 2 Calibration curves of the nomogram to predict DFS probability at 1-year (A and D), 3-year (B and E), and 5-year (C and F) in the training cohort and validation cohort.**

## DISCUSSION

In this single-centre retrospective study, the 30-day and 90-day nononcologic mortality rates of all 1,261 patients after pneumonectomy for primary lung cancer were 1.4% and 2.1%, respectively, and the estimated 5-year OS and DFS rates of the whole cohort were 51.6% and 66.1%, respectively. A total of 24 variables, including routine clinical, pathological, staging and treatment information, were included to select significant risk factors through Kaplan-Meier univariate analysis and further LASSO-Cox multivariate analysis. Then, DFS and OS nomogram models were developed to predict individualized survival probabilities. In the training and validation cohorts, all of the models calibrated well and demonstrated good to moderate predictive discrimination. In addition, the risk stratification models could bring more clinical benefit than the traditional TNM staging system. Most importantly, the web interactive calculators can be freely used at https://thoracicsurgery-nccchina.shinyapps.io/Disease-free-survival/ and https://thoracicsurgery-nccchina.shinyapps.io/Overall-survival/. By inputting the easy-to-available predictive variables (Fig. 5), individualized prediction of survival plot and probability (with 95% CIs) would assist thoracic surgeons or patients in making clinical decisions.

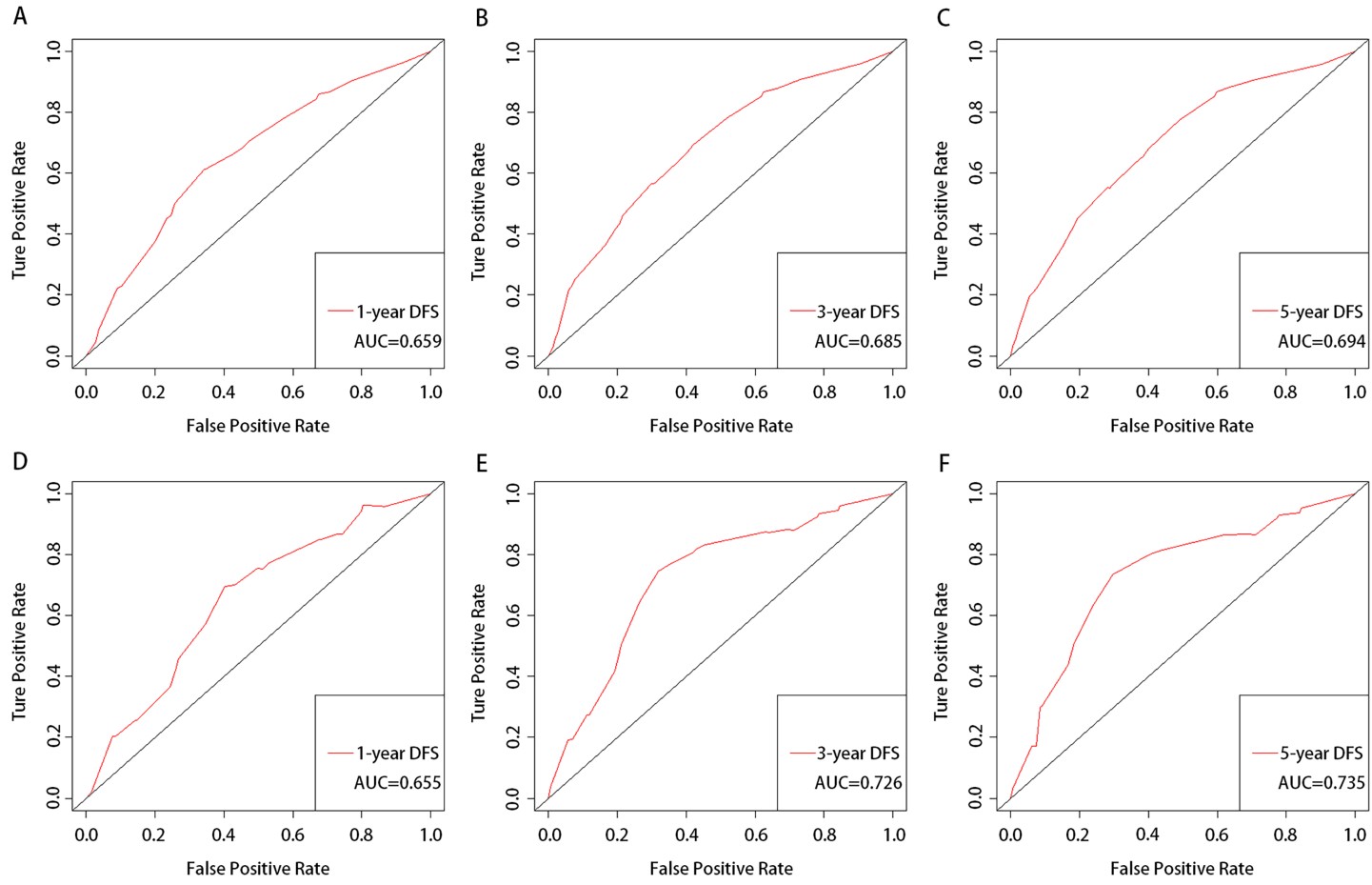

**Figure 3 Receiver-operator characteristic curve testing the power of the nomogram for predicting 1-year (A and D), 3-year (B and E), and 5-year (C and F) DFS probability in the training cohort and validation cohort.** AUC, area under the curve.

With the aid of the National Medicare Claims Database and the Nationwide Inpatient Sample, *Birkmeyer et al. (2002)* reported that the adjusted postpneumonectomy mortality rates in very high-volume hospitals (average no. of pneumonectomy/year >46 cases) were significantly lower than those in very low-volume hospitals (<9 cases). In our division, the annual cases of pneumonectomy remained stable over the past decade (median: 121, range: 94–129), which was more than that in any other report; therefore, plenty of clinical experience in preoperative (induction therapy, nutrition support, *etc.*), intraoperative (pulmonary artery pressure (PAP) and central venous pressure (CVP) monitoring, bronchial stump coverage or reinforcement, *etc.*), and postoperative management (liquid volume control, cardioversion, enhanced recovery after surgery, adjuvant therapy, *etc.*), were accumulated to reduce postoperative complications. Correspondingly, the 30-day and 90-day nononcologic mortality rates in this present large cohort were lower than those of previous studies (30-day mortality: 0–26.0%; 90-day mortality: 3.0–21.0%) (*Brunswicker et al., 2022*; *Tabutin et al., 2012*; *Yu et al., 2021*; *Yun et al., 2022*). Moreover, the long-term outcome was higher than that in the population-based analysis (*Yu et al., 2021*) and was

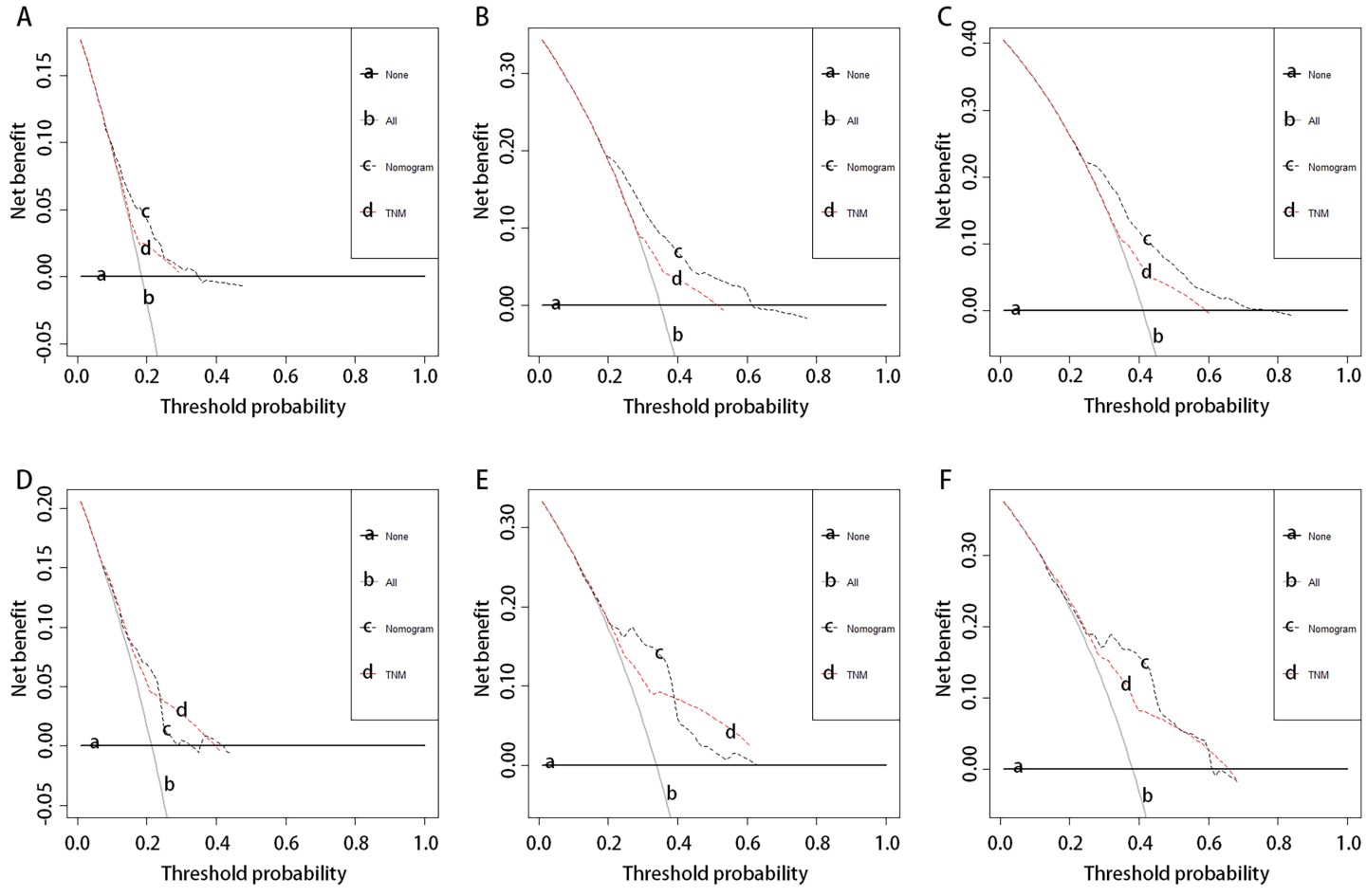

**Figure 4 Decision curves for the nomogram and the TNM staging system to predict 1-year (A and D), 3-year (B and E), and 5-year (C and F) DFS probability in the training cohort and validation cohort.** TNM, tumour-node-metastasis.

consistent with that in more recently published results (*Brunswicker et al., 2022*; *Yun et al., 2022*).

Primary tumour and/or metastatic lymph node invasion of pulmonary vessels and/or pericardium could provoke the spread of tumour cells (TCs) into the peripheral blood circulation, which leads to early distant metastasis and potential micrometastasis after surgery (*Wei et al., 2019*). Numerous studies have revealed that patients with main vessel or pericardium invasion had worse survival than patients with the same TNM staging (*Rami-Porta et al., 2015*; *Wei et al., 2019*). A total of 15.2% of patients in the current study, enough lengths could not be separated or the safety margins could not be ensured of the main pulmonary artery and/or vein due to invasion into the main pulmonary vessels and/or pericardium, and intrapericardial pneumonectomy was carried out. Consistent with other studies, intrapericardial ligation of the pulmonary vessels, especially arteries, reflected the potential release of TCs into the bloodstream, and intrapericardial artery ligation was notably associated with earlier disease recurrence and poorer prognosis in the multivariate analysis. A randomized clinical trial reported by *Wei et al. (2019)* indicated that the ligation of arteries prior to veins during lung cancer surgery was a significant risk

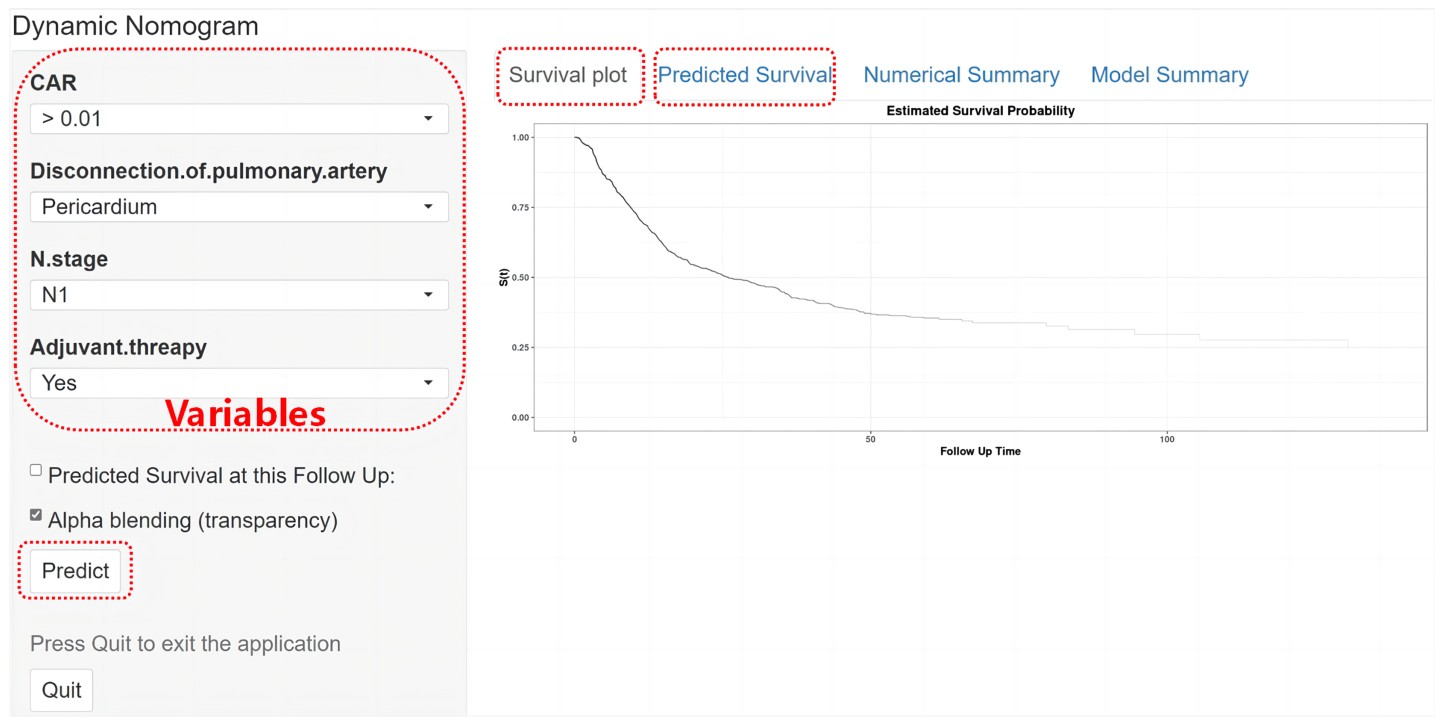

**Figure 5 The interface of our web-based dynamic nomogram for predicting DFS probability (with 95% CI) among patients who underwent pneumonectomy for primary lung cancer.** CAR, C-reactive protein to albumin ratio.

factor for increased circulating tumour cells (CTCs) in peripheral blood and was statistically linked to poorer long-term survival. Therefore, the pulmonary vein-first procedure should also be recommended in patients who undergo pneumonectomy (especially intrapericardial pneumonectomy) for primary lung cancer to reduce the risk of TCs directly entering the systemic circulation. In addition, surgical manipulation may cause the haematogenic dissemination of TCs, and therefore, no-touch isolation techniques should be reinforced during pneumonectomy to avoid potential iatrogenic TCs shedding (*Wei et al., 2019*).

Whether lung cancer patients after pneumonectomy can tolerant and benefit from adjuvant treatment is another controversial issue. A French multicentre retrospective study enrolled 1,466 patients who underwent pneumonectomy for non-small cell lung cancer (NSCLC) and reported that adjuvant treatment had no impact on long-term outcome (*Riquet et al., 2014*). In contrast, our present study suggested that postpneumonectomy treatment could significantly improve long-term survival; nevertheless, postpneumonectomy treatment due to advanced staging did not change the high recurrence rate. The cause of the different effects of postpneumonectomy treatment on survival and recurrence in this real-world cohort study was speculated to be that chemotherapy alone was selected as the main postpneumonectomy treatment regimen in most of the patients (530 of 544 patients, 88.8%). Postoperative chemotherapy can effectively prevent distant metastasis and thereby prolong the survival period in patients with NSCLC (*Pignon et al., 2008*); however, postoperative concurrent radiotherapy can

simultaneously reduce local recurrence risk (*Hui et al., 2021*). Similarly, *Hui et al. (2021)* also recently reported that, compared with postoperative chemotherapy alone, concurrent chemoradiotherapy for patients with pIII(N2) NSCLC after pneumonectomy could not only significantly reduce local recurrence and distant metastasis, but also improve DFS and OS (*Wang et al., 2019*). Therefore, we support that postpneumonectomy concurrent chemoradiotherapy should be recommended for locally advanced NSCLC patients who went through the perioperative period safely.

Compared with published studies regarding the prognosis of pneumonectomy (*Brunswicker et al., 2022*; *Riquet et al., 2014*; *Rivera et al., 2014*; *Tabutin et al., 2012*; *Wang et al., 2019*), there were four main advantages in the present study. First, this retrospective study was carried out based on a larger single-centre cohort, thus ensuring more homogenous diagnosis, treatment, and perioperative management. Second, almost all clinical, pathological, staging and treatment variables were included to screen for prognostic factors. Moreover, the LASSO regression model was applied for further predictor selection, and this model is less likely to be overfitted and can be more accurate than stepwise selection in the Cox proportional hazards model. Third, to our knowledge, this is the first integrated nomogram specifically for patients who underwent pneumonectomy for primary lung cancer to estimate prognosis. Fourth, the pictorial nomogram is contained in a website-based calculator, where easy-to-available variables are entered into and the likelihood of personalized survival is computed. We should acknowledge that the retrospective nature of our study inevitably resulted in several limitations. First, some patients were excluded as a result of missing data (*e.g.*, follow-up outcome), which may bring potential selection bias. Second, the time span of this retrospective cohort study was 11 years. During that period, the surgical techniques (robot-assisted thoracic surgery, sleeve lobectomy, *etc.*), incision methods (uniport and subxiphoid VATS, *etc.*), neoadjuvant (immunotherapy plus chemotherapy, targeted therapy, *etc.*), and adjuvant strategies (targeted therapy, immunotherapy, *etc.*), and the use of liquid biopsy for therapy monitoring (CTCs, minimal residual disease, *etc.*), had changed dramatically, and therefore, potential selection bias and follow-up bias were unavoidable. In addition, the nomogram was developed based on the supposition that all future endpoint events would be identical to the time of the patient enrolment; in other words, the predictive variables and accuracy of a nomogram were not to updated over time. Moreover, our nomograms were built and validated using single-centre data, and thus, whether the two nomograms can be universally used remains to be determined by validating it in an external population or a prospective cohort.

## CONCLUSIONS

In summary, we built two web-based interactive nomograms with good calibration and discrimination for individually predicting the DFS and OS of patients who underwent pneumonectomy for primary lung cancer. Moreover, our nomograms used for risk stratification could not only add clinical benefit to the traditional TNM classification system, but could also assist thoracic surgeons or patients in making personalized therapeutic recommendations and follow-up regimens.

## ACKNOWLEDGEMENTS

Presented at the 30th meeting of the European Society of Thoracic Surgeons (ESTS), Hague, Netherlands, June 19–21, 2022.

### Funding

This study was funded by the Shenzhen Key Medical Discipline Construction Fund (No. SZXK075) and the Sanming Project of Medicine in Shenzhen (No. SZSM201612097). The funders had no role in study design, data collection and analysis, decision to publish, or preparation of the manuscript.

### Grant Disclosures

The following grant information was disclosed by the authors:
Shenzhen Key Medical Discipline Construction Fund: SZXK075.
Sanming Project of Medicine in Shenzhen: SZSM201612097.

### Competing Interests

The authors declare that they have no competing interests.

### Author Contributions

- Xiangyang Yu conceived and designed the experiments, performed the experiments, analyzed the data, prepared figures and/or tables, authored or reviewed drafts of the article, and approved the final draft.
- Feng Wang performed the experiments, prepared figures and/or tables, authored or reviewed drafts of the article, and approved the final draft.
- Longjun Yang performed the experiments, prepared figures and/or tables, authored or reviewed drafts of the article, and approved the final draft.
- Kai Ma analyzed the data, authored or reviewed drafts of the article, and approved the final draft.
- Xiaotong Guo analyzed the data, authored or reviewed drafts of the article, and approved the final draft.
- Lixu Wang analyzed the data, authored or reviewed drafts of the article, and approved the final draft.
- Longde Du analyzed the data, prepared figures and/or tables, and approved the final draft.
- Xin Yu analyzed the data, prepared figures and/or tables, and approved the final draft.
- Shengcheng Lin analyzed the data, prepared figures and/or tables, and approved the final draft.
- Hua Xiao analyzed the data, prepared figures and/or tables, and approved the final draft.
- Zhilin Sui analyzed the data, prepared figures and/or tables, and approved the final draft.
- Lanjun Zhang conceived and designed the experiments, authored or reviewed drafts of the article, and approved the final draft.

- Zhentao Yu conceived and designed the experiments, authored or reviewed drafts of the article, and approved the final draft.

## Human Ethics

The following information was supplied relating to ethical approvals (*i.e.*, approving body and any reference numbers):

The institutional ethics committee at Sun Yat-sen University Cancer Center approved this retrospective study (No. B2022-011-01).

## Data Availability

The raw data are available in the Supplemental File.

## Supplemental Information

Supplemental information for this article can be found online at http://dx.doi.org/10.7717/peerj.15938#supplemental-information.

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
