# Peer review of "Development and validation of web-based dynamic nomograms predictive of disease-free and overall survival in patients who underwent pneumonectomy for primary lung cancer"

_PeerJ, doi:10.7717/peerj.15938_

## Round 0.1 · original submission · Minor Revisions

Please revise the manuscript as suggested.

Reviewer 1 ·

Basic reporting

no comment

Experimental design

no comment

Validity of the findings

no comment

Additional comments

I have read and reviewed the manuscript “Development and validation of web-based dynamic nomograms predictive of disease-free and overall survival in patients who underwent pneumonectomy for primary lung cancer”. The general idea of a nomogram that takes into account the T and N elements along with some of the details about why pneumonectomies are performed to help predict both OS and DFS has interest. However, much work should be done to improve this manuscript before publication. I have the following comments:
1. In the "Methods" section, please clarify the definition of a positive margin.
2. In the "Methods" section, please specify how many patients adhered to the follow-up strategy and if there were routine procedures for detecting brain or bone metastasis.
5. In the "Results" section, adjuvant therapy is employed as a survival predictor. Please provide the indications for adjuvant therapy and address the concern that patients who did not receive adjuvant therapy may have had worse conditions, such as increased comorbidity, leading to poorer survival outcomes.
7. In the "Results" section, the AUCs of the nomograms mostly range from 0.6 to 0.7, which are unsatisfactory.
8. In the "Results" section, it would be interesting to know if there are outcome differences between right and left pneumonectomy.

·

Basic reporting

Clear and logical. Appropriate presentation.

Experimental design

Well designed.

Validity of the findings

The finding is validated properly.

Reviewer 3 ·

Basic reporting

no comment

Experimental design

no comment

Validity of the findings

no comment

Additional comments

In this study, Yu and colleagues identified the independent clinicopathological variables that predict disease-free and overall survival in patients who underwent pneumonectomy for primary pulmonary malignancy based on a real-world cohort analysis. Moreover, web-based servers according to the integrated nomogram models have been developed and are freely available for thoracic surgeons to input the predictive variables required for the individualized DFS and OS probability. I only have some minore comments: a) The English and grammar should be checked throughout the whole paper. b) How many patients were excluded according to the exclusion criteria? I recommend the authors take a figure to show the screening process. c) The clinical implications of the two nomograms remain unclear and require further clarification.

---

## Round 0.2 · accepted · Accept

This manuscript can be accepted.

Reviewer 3 ·

Basic reporting

no comment

Experimental design

no comment

Validity of the findings

no comment

Additional comments

The revised manuscript has addressed all of the previous concerns. The authors have done a commendable job in addressing the issues raised and improving the overall quality of the manuscript. I strongly recommend the acceptance of this manuscript for publication.